# Protective Properties of Marine Alkyl Glycerol Ethers in Chronic Stress

**DOI:** 10.3390/md21040202

**Published:** 2023-03-24

**Authors:** Ruslan M. Sultanov, Tatiana S. Poleshchuk, Ekaterina V. Ermolenko, Sergey P. Kasyanov

**Affiliations:** 1A.V. Zhirmunsky National Scientific Center of Marine Biology, Far Eastern Branch, Russian Academy of Sciences, 17 Palchevskogo Str., Vladivostok 690041, Russia; 2Faculty of Pediatrics and Pharmacy, Pacific State Medical University, 2 Ostryakova Ave., Vladivostok 690002, Russia

**Keywords:** alkyl glycerol ethers, chronic stress, plasmalogens, chimyl alcohol, batyl alcohol, immunity, adaptogenic action

## Abstract

In this paper we discuss the effect of alkyl glycerol ethers (AGs) from the squid *Berryteuthis magister* on a chronic stress model in rats. The study was performed on 32 male Wistar rats. Animals received AGs at a dose of 200 mg/kg through a gavage for six weeks (1.5 months), and were divided into four groups: group 1 (control), group 2 (animals received AGs), group 3 (stress control), group 4 (animals received AGs and were subjected to stress). Chronic immobilization stress was induced by placing each rat into an individual plexiglass cages for 2 h daily for 15 days. The serum lipid spectrum was evaluated by the content of total cholesterol, triglycerides, high-density lipoprotein cholesterol, low lipoprotein cholesterol and very low-density lipoprotein cholesterol. The atherogenic coefficient was calculated. The hematological parameters of peripheral blood were evaluated. The neutrophil–lymphocyte ratio was counted. The levels of cortisol and testosterone in blood plasma were determined. AGs at the selected dose did not have a significant effect on the body weight of rats in the preliminary period of the experiment. Under stress, the body weight gain, the concentrations of very low-density lipoprotein cholesterol and blood triglycerides decreased significantly. The neutrophil–lymphocyte ratio in animals treated with AGs shifted towards lymphocytes. A favorable increase in the percentage of lymphocytes was found in the stressed group of animals treated with AGs. So, for the first time, it was found that AGs prevent stress-induced suppression of the immune system. This confirms the benefit of AGs for the immune system under chronic stress. Our results prove the efficiency of the use of AGs for treating chronic stress, a serious social problem in modern society.

## 1. Introduction

From the moment of its first introduction to the present, the problem of stress and adaptation to stress has not lost its relevance. The social and domestic environment of modern life creates long-term, constant psycho-emotional stress in the vast majority of people. Constant mental and psycho-emotional stress and contravention of the regime of work, rest and nutrition often lead to the breakdown of adaptation mechanisms and the development of diseases. Selye [1] provided a long list of diseases of adaptation, which includes diseases of the cardiovascular system (such as cardiac diseases, hypertension, hypotension, arteriosclerosis and other), the nervous system (e.g., psychosomatic derangements, neuroses, neurasthenia, psychoses), the gastrointestinal tract (in particular, gastrointestinal ulcers and ulcerative colitis), anemias, and other disorders.

More and more substances are under investigation as potential treatment to reduce stress and its consequences. The purpose of our work is to determine whether the use of AGs is an effective treatment for chronic stress.

AGs are precursors of such biologically active substances as plasmalogens and the platelet-activating factor (PAF). The PAF is a powerful bioregulator with a broad spectrum of physiological functions. PAF has gained major attention as a versatile inflammatory mediator produced by different types of immune cell, in particular neutrophils, eosinophils and macrophages, but also endothelial cells and platelets [2].

Plasmalogens are a class of membrane glycerophospholipids with unique properties. Plasmalogens are critical for human health; they are important for neuronal development, for immune response and as endogenous antioxidants. Given the abundance and distribution of plasmalogens in multiple membrane systems and in different tissues, it is not surprising that plasmalogens play a key role in multiple biological functions [2,3,4]. There is growing evidence of the therapeutic potential of modulating plasmalogen levels to prevent or treat cardiometabolic and neurodegenerative diseases [3,4].

Previously, our team conducted a study of the AGs influence in aquite immobilization stress, which revealed their positive effect [5]. This study was a starting point for our interest in this problem. There are no other works studying AGs under stress conditions.

## 2. Results

### 2.1. Changes in Body Weight in Rats Treated with AGs

At the preliminary stage of the experiment (treatment with AGs before the onset of stress exposure), there were no significant differences (t = 0.3; *p* = 0.76) in the general body weight gain between animals treated with AGs (general body weight gain 23.2 ± 5.6%) and animals in the control group (general body weight gain 22.5 ± 4%) (Figure 1).

### 2.2. Body Weight Changes under Stress

Significant differences in body weight gain were found after the first week of stress (H = 9.0; *p* = 0.01) as well as at the end of the experiment (H = 9.3; *p* = 0.01). Stress expectedly decreased body weight with the strongest effect in the first week (see Appendix A and Appendix B). In the second and third weeks of stress, differences in body weight gain between stressed animals and animals treated with AGs against the background of stress and without it were no longer significant.

### 2.3. Content of Lipoproteins in Blood Serum

Table 1 shows the parameters of lipid metabolism in rats after the end of the experiment. An analysis of variance (H = 9.24; *p* = 0.0262) revealed differences between groups in triglyceride concentrations (Figure 2). Treatment with stress and AGs leads to a decrease in triglyceride levels. There were significant differences between control rats and stressed animals (t = 2.7, *p* = 0.03), as well as between rats treated with AGs and subjected to stress (t = 2.9, *p* = 0.02), but not between the control group and the group treated with AGs (t = 1.33; *p* = 0.22), or between stressed animals and those treated with AGs before stress (t = −0.1; *p* = 0.91).

In addition, using an analysis of variance, we determined differences (H = 9.68; *p* = 0.0215) among rats in the concentration of VLDL-C (Figure 3). Exposure to stress (t = 2.69, *p* = 0.03) and stress with AGs (t = 2.96, *p* = 0.02) lead to a decrease in VLDL-C levels. Treatment with AGs decreased VLDL-C levels, but insignificantly (t = 1.31, *p* = 0.23) compared to the control group. Additionally, treatment with AGs before stress only insignificantly increased VLDL-C levels (t = 2.0, *p* = 0.1). So, only stress influenced VLDL-C levels significantly.

Thus, the amount of VLDL-C and blood triglycerides significantly decreased under stress.

### 2.4. Effect of AGs on Content and Composition of Fatty Acids and Plasmalogen Levels in Plasma Lipids

The fatty acid composition of the plasma total lipids of rats is presented in Appendix C. Significant differences in the composition of the main fatty acids were not found. Figure 4 showed a change in the level of plasmalogens in the ratio 16:0 DMA/16:0 FAME in experimental groups. Under the action of AGs (*p* < 0.05), the concentration of 16:0 DMA increased in groups with oral administration of AGs compared with control animals.

### 2.5. Blood Test Results

The hematological parameters of the clinical blood tests of the animals are given in Appendix D. No significant differences were found between the groups in terms of the number of erythrocytes (H = 2.07; *p* = 0.56), the hematocrit level (H = 1.63; *p* = 0.65) and the amount of hemoglobin (H = 4.02; *p* = 0.26).

There was a significant difference between the groups in terms of lymphocytes percentage (H = 8.49; *p* = 0.037) (Figure 5). The Mann–Whitney test revealed differences between the control group and the animals treated with AGs against the background of stress (Z = −2.08; *p* = 0.03) and between the stress-control group and the animals treated with AGs against a background of stress (Z = −2.33; *p* = 0.016). So, the percentage of lymphocytes decreased under stress action and vice versa increased in the stress group treated with AGs.

The neutrophil–lymphocyte ratio also differed between the groups (H = 9.15; *p* = 0.0273) (Figure 6). In animals subjected to chronic stress, the ratio was higher than in the group treated with AGs against the background of stress (Z = 2.33; *p* = 0.016), and higher than in the control group.

### 2.6. Morphometry

The mass indices of the adrenal gland, spleen, and thymus are given in Table 2.

The Kruskal–Wallis test could not reveal significant differences between the groups, based on the mass indices of the spleen (H = 1.65 *p* = 0.65), thymus (H = 2.63 *p* = 0.45) and adrenal gland (H = 3.11; *p* = 0.38).

### 2.7. Hormone Levels

Table 3 and shows the levels of testosterone (H = 0.79; *p* = 0.85) and cortisol (H = 3.79; *p* = 0.28).

There were no significant differences in the levels of these hormones between the groups. Nevertheless, the maximum level of testosterone was determined in animals treated with AGs.

## 3. Discussion

Immobilization is generally extensively used as a stressor for the study of stress-related biological, biochemical and physiological responses in animals. Immobilization can be produced in two different ways. Animals can be kept immobilized in a semi-cylindrical acrylic tube with proper holes in it for air to pass, or animals can be stretched on a board with the limbs immobilized with adhesive tape. In the second case, movements of head are restricted with a metal loop coiled around the neck [6]. There are also different immobilization regimens detailed in the current literature. We chose a classical stress model for our study [7,8,9]. Even mere emotional stress can lead to a severe alarmed reaction. Nervous commotions are extremely effective in alarming stimuli. Through its complex ramification, the nervous system readily transmits impulses to many parts of the body, and if these stimuli are strong, they cause systemic stress [1].

AGs in the selected dose do not produce a significant effect on the general body weight gain of rats after 1.5 months of application. According to the literature, high-dose selachyl alcohol decreased the body weight, serum triglycerides, cholesterol, fasting glucose levels, insulin levels, and serum leptin levels of the high-fat diet-fed mice, while high-dose batyl alcohol increased the fasting insulin levels of the high-fat diet-fed mice [10].

During the experiment, spleen samples were stored. Then, some of the samples were used to assess the level of plasmalogens in this organ. The conclusion was that the spleen was weakly affected by chronic immobilization stress; fluctuations in the content of fatty acids were insignificant. Thus, no influence of AGs on the spleen lipid composition was revealed [11].

Chronic stress led to a decrease in animal body weight during first week. Thus, the most significant effect of stress was observed in the first week of immobilization. The absence of significant changes in the mass of the thymus, adrenal glands and spleen indicates the development of the resistance stage. This is also indicated by the absence of pronounced differences in cortisol levels. In addition, according to the literature, an increase in the level of corticosterone characterizes acute, but not chronic, homotypic stress [7,12].

Some very significant data of our study are the percentage of lymphocytes and neutrophil–lymphocyte ratios. Elevated neutrophil–lymphocyte ratios were associated with chronic stress exposure [12]. Based on cortisol levels and neutrophil–lymphocyte ratios together, we can conclude that the model of chronic immobilization stress has been successfully implemented.

The increase in the percentage of lymphocytes in the stressed group of animals during treatment with AGs is consistent with the literature data. AGs can modulate immune responses by boosting the proliferation and maturation of murine lymphocytes in vitro [13]. Mice deficient in the peroxisomal enzyme glyceronephosphate O-acyltransferase (GNPAT), essential for the synthesis of ether lipids, had significant alteration of the thymic maturation of iNKT cells and fewer iNKT cells in both the thymus and peripheral organs [14]. Shark liver oil is able to increase lymphocyte proliferation in rats [15], leucocytes, lymphocytes, neutrophils, monocytes and IgG level in blood of piglets from supplemented sows [16]. White blood cells, IgG and lymphocytes significantly increased, whereas neutrophils significantly decreased in aging patients after surgical treatment thanks to AGs in a dose of 500 mg twice a day for a duration of 4 weeks [17]. So, we can conclude that AGs have an indirect effect on blood cells [18].

We consider the increase in the percentage of lymphocytes in the stressed group of animals under the influence of AGs as a positive sign, since prolonged suppression of immunity during chronic stress is unfavorable for the body. Restoring plasmalogen levels through use of plasmalogen replacement therapy has been shown to be a successful anti-inflammatory strategy [19]. Consequently, animals treated with AGs were better able to tolerate chronic stress.

Thus, AGs have some beneficial properties in both acute [5] and chronic stress.

## 4. Materials and Methods

### 4.1. Animals

This study was carried out on 32 male Wistar rats (eight in each group) for 2 months. Animals were kept in vivarium, on a standard diet without any additions, with free access to food and water. The average weight of the animals at the beginning of the experiment was 171 ± 19 g. All procedures were approved by the Animal Ethics Committee at A.V. Zirmunsky National Scientific Center of Marine Biology (Far Eastern Branch, Russian Academy of Sciences, Vladivostok, Russia), according to the Laboratory Animal Welfare guidelines.

### 4.2. Preparations

#### 4.2.1. Preparation of the AGs

The squid *Berryteuthis magister* was fished in the Bering Sea during the year before our experiment. After squid processing, the liver was removed and stored for 3 months at −18 °C. All chemical reagents used in this study are of analytical grade (Sigma-Aldrich, St. Louis, MO, USA). All solvents were of HPLC grade, supplied by Sigma-Aldrich (USA).

Extraction of total lipids was conducted according to Bligh and Dyer [20]. Saponification of lipids was carried out according the procedure described by Christie [21].

Precipitation of AGs from saponified lipids was carried out by double crystallization in acetone at different temperatures [22]. The amount of AGs in the mixture was determined by GC using the AG-C8:0 standard and using TLC according to the procedure [22]. The final form is used in the supplement “NanoMind” (Moscow, Russia).

#### 4.2.2. Determination of AGs Composition

Composition of AGs as TMS-AGs was determined by GC and GC-MS.

TMS-AG were prepared as follows: a volume of 50 μL of *N*,*O*-Bis(trimethylsilyl)-trifluoroacetamide was added to 5 mg AGs; the mixture was heated to 80 °C for 1 h. After addition of 200 μL of hexane, 1 μL of each silylated fraction was injected into the GC. The composition of TMS-AGs was determined by GC using a Shimadzu GC-2010 Plus chromatograph with a flame ionization detector (Kyoto, Japan) and a Supelco SLB™-5 ms capillary column, 30 m × 0.25 mm I.D. (USA). Separation of mixture components was carried out under the following conditions: an initial temperature 200 °C and a heating rate of 2 °C/min to 290 °C. The temperature was held for 35 min. The injector temperature was 270 °C and the detector temperature was 250 °C. AGs identification was performed by comparison with the available known standards. GC-MS was used to identify the structure of TMS-AGs. The electronic impact spectra were recorded using a Shimadzu TQ-8040 (Kyoto, Japan) gas chromatography-mass spectrometer (with the column Supelco SLB™-5 ms at 70 eV, under the same temperature conditions as during GC.

The composition of the precipitated saturated fraction of AGs is given in Table 4, and the chemical structure image in Figure 7.

Despite the fact that the content of unsaturated AGs in the initial hydrolyzed lipids was 17.5% (of the total amount of AGs), the resulting preparation contained only saturated radicals.

The resulting AGs’ preparation was white, odorless friable powder.

### 4.3. Biological Experiment

#### 4.3.1. Animal Treatment

We used a preparation of 99% AGs with a chemical alcohol content (radical 16.0) which was 95% of the total AGs (Table 4 and Figure 7). The AGs intended for use in animals were dispersed in water. Rats received medicine or water orally through a catheter for six weeks (1.5 months).

Experimental groups (eight animals in each):

Group 1—Animals received water (control);

Group 2—Animals received AGs, 200 mg/kg of body weight;

Group 3—Animals received water and were subject to stress (stress control);

Group 4—Animals received AGs at a dose of 200 mg/kg and were subjected to stress.

Gavage administration of various vehicles can result in aspiration, pulmonary injury, and/or elicitation of a stress response in a vehicle- and dose volume-dependent fashion. The dose volumes for gavage administration in the rat generally should not exceed 10 mL/kg. Gavage administration of corn oil at ≥20 mL/kg, but not 1% methylcellulose/0.2% Tween 80 or water, induced a stress response in a volume-dependent fashion, resulting in elevated plasma corticosterone levels [23]. In our study, the volume of the administered substance did not exceed 2 mL/kg. Therefore, we assume that this manipulation did not lead to a significant deviation in the condition of the animals in our study, so we did not introduce an additional intact control group into the study.

#### 4.3.2. Stress Procedure

Chronic immobilization stress [6] was induced in rats by placing the animals in individual plexiglass cases with holes for ventilation, for 2 h [7] daily for 15 days [8]. The regimen has analogues in the literature [9].

#### 4.3.3. Measurement of Body Weight

During the experiment, the animals were weighed with an electronic weighing balance at intervals of 5–10 days.

General body weight gain during AGs treatment before stress, general body weight gain during stress action and body weight gain during each week of stress were calculated as follows:Weight gain (%) = (W1 − W0)/W0

initial weight gain (W0)

new weight gain (W1)

#### 4.3.4. Collection of Samples

Rats were anesthetized with Isofluran and sacrificed by decapitation. Blood for biochemical and hematological studies was taken at decapitation.

The adrenal glands, spleen and thymus were weighed.

#### 4.3.5. The Serum Lipid Spectrum of Rats

The serum lipid spectrum was evaluated by the contents of total cholesterol (TC), triglycerides (TG), high-density lipoprotein cholesterol (HDL-C), low lipoprotein cholesterol (LDL-C) and very low-density cholesterol (VLDL-C). The TC and TG concentrations were determined by the enzymatic colorimetric method on a ChemWell Combo 2910 automatic biochemical analyzer (Awareness Technology, Palm City, USA), and the concentration of HDL-C—on a RIELE 5010V5 + biochemical semi-automatic photometer (Riele, Berlin, Germany) with the use of Olveks Diagnosticum kits (Sankt-Peterburg, Russia). LDL-C and VLDL-C were calculated by the formula of Friedwald [24,25] in modification (see example in [26]).
LDL-C = TC − VLDL-C − HDL-C 
VLDL-C = TG/2 

The results were expressed in μm/L.

The atherogenic coefficient (AC) was calculated by formula [27]:AC = (TC − HDL-C)/HDL-C 

#### 4.3.6. The Content of Fatty Acids and Dimethyl Acetals (DMAs) in the Total Plasma Lipids of Rats

Lipids from rat plasma were extracted by the Bligh and Dyer method, 1957 [20]. Methyl esters of fatty acids (FAMEs) and DMAs from liver lipids of rats were prepared according to the procedure of Carreau and Dubacq, 1978 [28]. The α,β-unsaturated ether in the plasmalogen molecule was converted into a DMA of the corresponding aldehyde during transesterification, and the relative amount of plasmalogen was thus reflected in the ratio between 18:0 aldehyde and stearic acid, as well as by the ratio between 16:0 aldehyde and palmitic acid according Bjorkhem et al., 1986 [29]. An analysis of FAMEs and DMAs was conducted by GC on a Shimadzu GC-17A chromatograph (Shimadzu) with a flame ionization detector and a capillary column 30 m × 0.25 mm, i.d. Supelcowax 10 (Bellefonte). An analysis was performed under the following conditions: a column temperature of 190 °C, an injector, and a detector temperature of 240 °C. Helium was used as a carrier gas. The peaks of methyl esters of fatty acids were identified by retention times of individual FA esters through comparison of their equivalent carbon length numbers with authentic standards (PUFA-3mix from menhaden oil was purchased from Supelco). DMA identification was carried out by comparison of retention times with standards of 16:0 DMA and 18:0 DMA (Sigma-Aldrich).

#### 4.3.7. Hormone Research

Plasma cortisol and testosterone levels were determined by immunoassays using Vector kits (Novosibirsk, Russia) according to the manufacturer’s method.

#### 4.3.8. Hematological Analysis

Hematological parameters of the peripheral blood, including hematocrit, trombocrit, the number of red blood cells, platelets, white blood cells, lymphocytes, neutrophils eosinophils, basophils and monocytes (MID), the mean volume of erythrocytes, the coefficient of variation of the range of erythrocyte distribution, the standard deviation of the range of erythrocyte distribution, the mean platelet volume and the range of platelet distribution, the level of hemoglobin in the blood, the mean concentration of hemoglobin in erythrocytes and the average hemoglobin content in one red blood cell were evaluated with the use of an Abacus Hematology Analyzer (Diatron, Medley, FL, USA).

The neutrophil–lymphocyte ratio was calculated [12].

### 4.4. Statistical Analysis

The Shapiro–Wilk test was used to verify the distribution of data. The data are presented as the mean ± standard deviation or as median, lower (25%, Q25) and upper (75%, Q75) quartiles. Statistical analysis of the data was performed with the program «Statistica 10.0». A one-way ANOVA, two-sided unpaired Student’s *t*-test, Kruskal–Wallis test and Mann–Whitney test were used to identify statistically significant differences. A statistical probability of *p* < 0.05 was considered significant [30].

## 5. Conclusions

AGs prevent many of the negative effects of acute and chronic stress. In particular, they improve the immune status of animals under chronic stress.Preparations with the participation of AGs can be useful in the prevention and treatment of socially significant diseases of today’s world.

## Figures and Tables

**Figure 1 marinedrugs-21-00202-f001:**
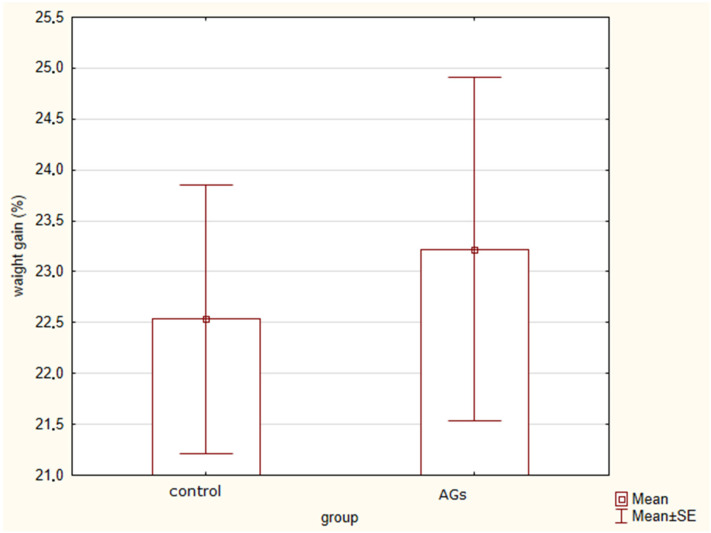
General body weight gain in rats treated with AGs, indicating no significant differences.

**Figure 2 marinedrugs-21-00202-f002:**
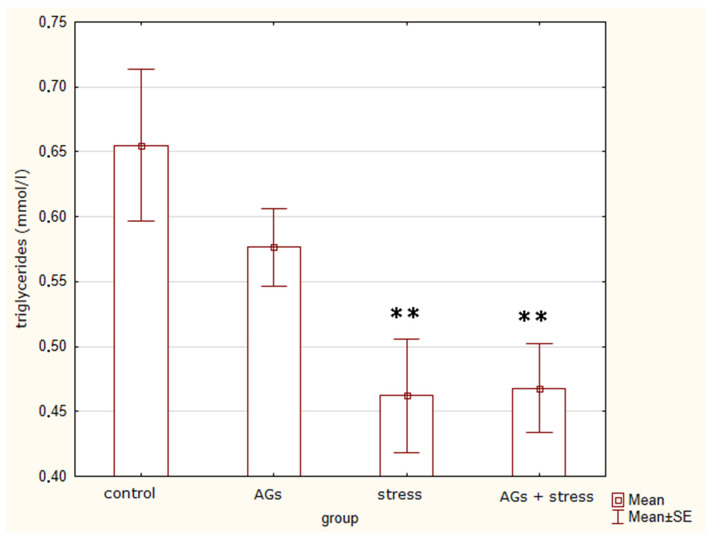
Triglycerides concentration, indicating statistically significant differences between the control group and experiment groups: ** *p* < 0.05.

**Figure 3 marinedrugs-21-00202-f003:**
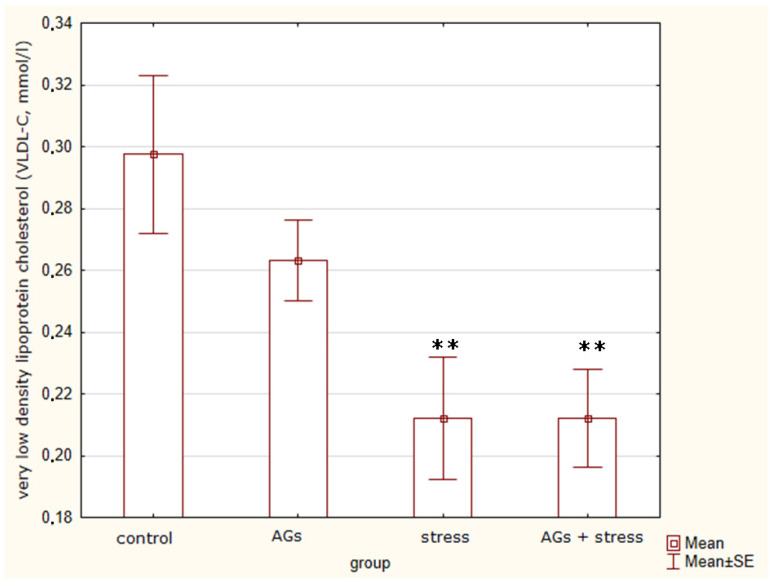
Very low-density lipoprotein cholesterol concentration, indicating statistically significant differences between the control group and experiment groups: ** *p* < 0.05.

**Figure 4 marinedrugs-21-00202-f004:**
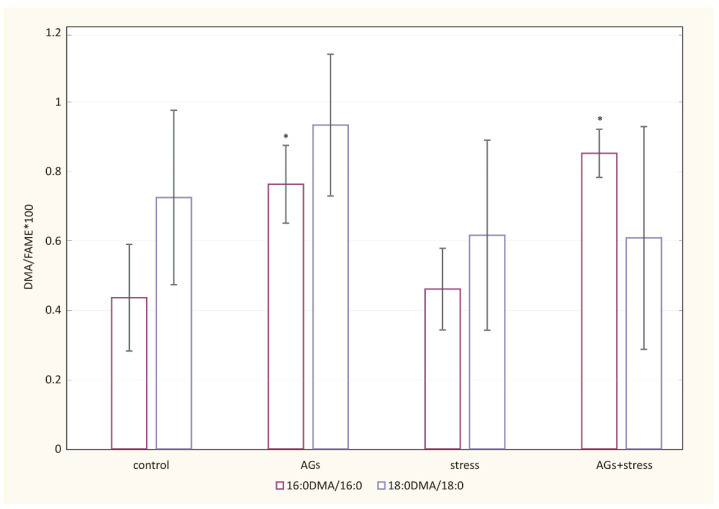
Effect of AGs on plasma lipids of control rats and experiment groups. The relative amount of plasmalogen was reflected in the ratio of 18:0 of dimethyl acetal (DMA) (% of total FAME and DMA) to stearic acid (% of total FAME and DMA), as well as by the ratio of 16:0 of DMA (% of total FAME and DMA) to palmitic acid (% of total FAME and DMA). * Statistically significant differences between the control group and experiment groups: * *p* < 0.05.

**Figure 5 marinedrugs-21-00202-f005:**
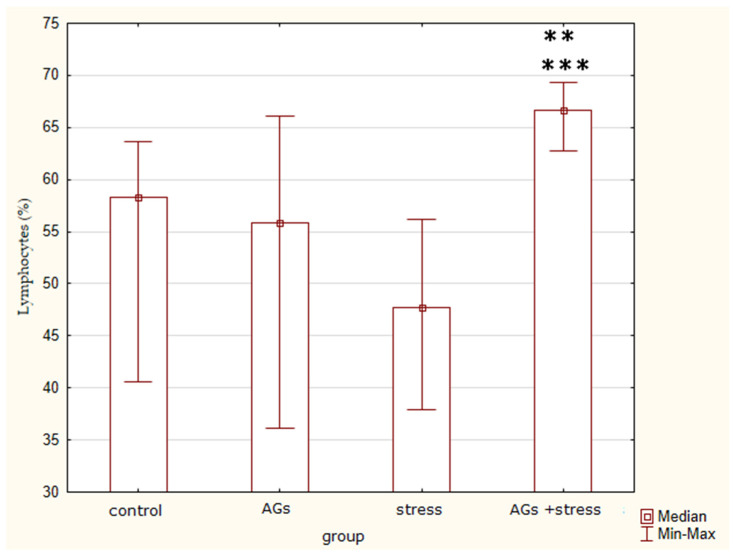
Lymphocytes percentage, indicating statistically significant differences in the control group ** and stress group *** *p* < 0.05.

**Figure 6 marinedrugs-21-00202-f006:**
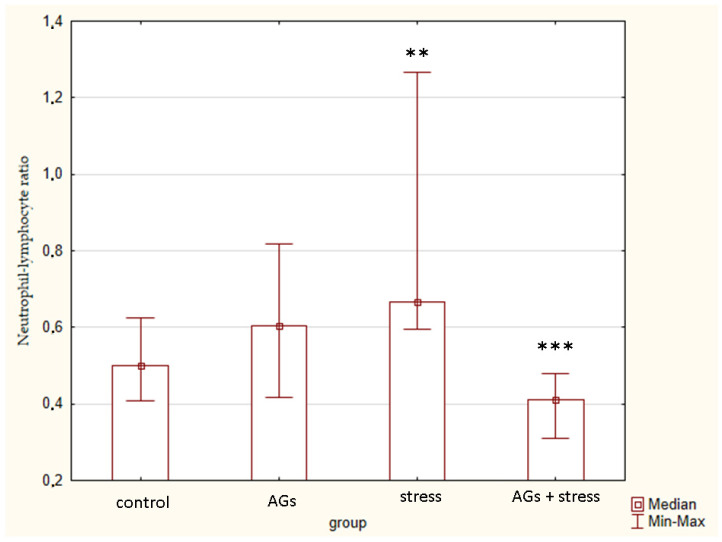
Neutrophil–lymphocyte ratio, indicating statistically significant differences in the control group ** and stress group *** *p* < 0.05.

**Figure 7 marinedrugs-21-00202-f007:**
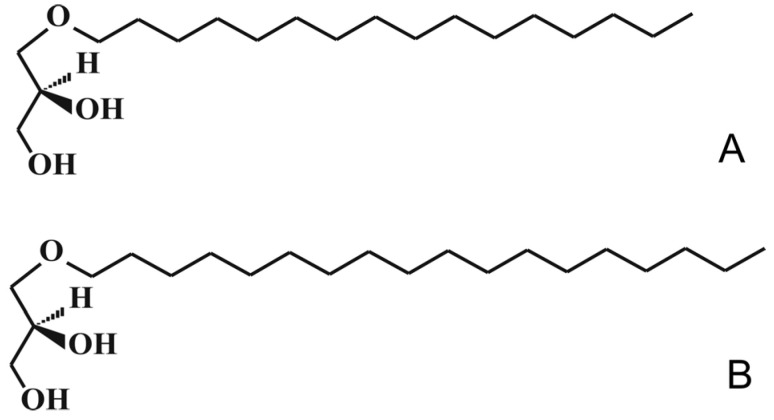
The chemical structure of alkylglycerols. (**A**)—chimyl alcohol, (**B**)—batyl alcohol.

**Table 1 marinedrugs-21-00202-t001:** Indicators of lipid metabolism.

Indicator	Control	AGs	Stress	AGs + Stress	
Total cholesterol, mmol/L	1.8 1.4–2.05	1.6 1.2–1.6	1.4 1.2–1.5	1.1 1.1–1.4	H = 6.64′ *p* = 0.08
HDL-C, mmol/L	0.88 ± 0.16	0.98 ± 0,16	0.93 ± 0.16	0.9 + 0.19	F = 0.34″ *p* = 0.79
LDL-C, mmol/L	0.55 0.24–0.85	0.18 0.1–0.45	0.15 0.06–0.26	0.1 0.05–0.11	H = 3.84 *p* = 0.28
VLDL-C, mmol/L	0.3 ± 0.05	0.26 ± 0.03	0.21 ± 0.04 **	0.21 ± 0.04 **	F = 4.9 *p* = 0.013 *
Atherogenic coefficient	0.89 0.65–1.34	0.43 0.36–0.7	0.34 0.33–0.56	0.45 0.22–0.47	H = 4.45 *p* = 0.22
Triglycerides, mmol/L	0.66 ± 0.12	0.58 ± 0.07	0.46 ± 0.1 **	0.47 ± 0.08 **	F = 4.8 *p* = 0.014 *

HDL-C—high-density lipoprotein cholesterol, low lipoprotein cholesterol (LDL-C) and very low-density lipoprotein cholesterol (VLDL-C). * significant differences; ** significant differences with control group; ′—Kruskal–Wallis test; ″—one-way ANOVA.

**Table 2 marinedrugs-21-00202-t002:** Organ weights.

Indicator	Control	AGs	Stress	AGs + Stress	
Spleen index	0.32 0.29–0.35	0.29 0.24–0.32	0.29 0.28–0.39	0.28 0.23–0.34	H = 1.65′ *p* = 0.65
Glandula adrenalis index	0.026 0.024–0.027	0.028 0.026–0.028	0.027 0.027–0.028	0.027 0.027–0.03	H = 3.11 *p* = 0.38
Thymus index	0.098 ± 0.02	0.1 ± 0.014	0.093 ± 0.037	0.089 ± 0.008	F = 0.26″ *p* = 0.85

′—Kruskal–Wallis test; ″—one-way ANOVA.

**Table 3 marinedrugs-21-00202-t003:** Hormone levels.

Indicator	Control	AGs	Stress	AGs + Stress	
Testosterone, nmol/L	12.6 11.8–18.3	11.7 8.2–16.9	24.4 7.65–31.6	16.5 11.8–24.7	H = 1.26′ *p* = 0.74
Cortisol, nmol/L	13.7 10.7–18.2	56.9 17.5–58.3	32.1 13.7–38.4	28.3 17.6–51	F = 1.4″ *p* = 0.28

′—Kruskal–Wallis test; ″—one-way ANOVA.

**Table 4 marinedrugs-21-00202-t004:** AGs composition after double crystallization.

Alkyl Chain	AGs Composition	Trivial Name
C14:0 ^a^	1.52 ± 0.15 ^b^	
C16:0	93.77 ± 0.16	Chimyl alcohol
C18:0	4.71 ± 0.25	Batyl alcohol
Σsat	100	
Σunsat	0	

sat—saturated AGs, unsat—unsaturated AGs; ^a^—indicated by chain length and double bond of alkyl chain in AGs; ^b^—mean weight % ± standard deviation (*n* = 5).

## Data Availability

Not applicable.

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
