# Peer review of "Protective Properties of Marine Alkyl Glycerol Ethers in Chronic Stress"

_marinedrugs, 2023, doi:10.3390/md21040202_

Round 1

Reviewer 1 Report (Previous Reviewer 2)

This paper describes the protective effect of marine alkylglycerol esters (AGs) on a rat chronic stress model. The authors revised their manuscript sincerely, but some points still need to be revised.

1) Poor preparation of Figures and Tables !!

1-1: Inconsistent display format in Figs 1-4 (box plot), Fig 5 (bar chart) and Figs 6-7 (box-and-whisker plot).

 1-2: Fig2: Why are the groups displayed in a different order?

 1-3: There is Table 5 but not Table 4.

 1-4: Appendix A and B have layout problems.

 2) The accuracy of the text should be checked !!

2-1: Ln12: “six week” should be “six weeks”.

2-2: Ln196: “[7, 8, 9” should be “[7, 8, 9]”.

2-3: Ln222: “AKGs” is probably “AGs”.

Author Response

Thank you for your comments. Answers are in the file below.

Reviewer 2 Report (Previous Reviewer 3)

The revised manuscript by Dr. Sultanov and colleagues describes the effects of dietary alkylglycerol supplementation on stress using a rat model. The authors have addressed most of my specific criticisms. There remains a large number of typographical errors and overall poor English usage that detracts from the manuscript.

Specific points:

1. The authors report repeatedly in the manuscript that the “number of lymphocytes” is increased in the stress + AG animals. In reality, it is the “percentage of lymphocytes” that increases. The actual number of lymphocytes is not increased (see Appendix C table).

2. Most of the data reported in the manuscript does not support the efficacy of AGs on stressed animals (as discussed in Discussion).  

3. How long does it take to reach a stress-resistant stage? Are the animals in the stress-resistant stage by 6 weeks?

4.. Misspelling errors: line 75,“weak” should be “week”; line 161, “limphosites” should be lymphocytes; Fig 7, “Limph” should be lymph; line 255, “Suponification” should be Saponification; line 273, “Electronic” should be Electron; line 276, “Table 5” should be Table 4; line 293, “Table 1” should be “Table 4”; line 312, “15 days” should this be 14 days? See line 313.

5. The column headings for Appendix B Table are erroneously placed in Appendix A.

6. Appendix, line 24: “An increase in the number of lymphocytes” should be changed to “A favorable increase in the percentage of lymphocytes”.

7. Line 195-196: “We chosen quite common and comfortable for our study”. What does this sentence mean?

Author Response

Thank you for your comments. Answers are in the file below. 

Reviewer 3 Report (New Reviewer)

The manuscript entitled " Protective properties of marine alkyl glycerol ethers in chronic stress" by Ruslan et al. studied the protective effect of marine alkyl glycerol ethers in chronic stress. The result suggested that proves the efficiency of using AGs in chronic stress, a severe social problem in modern society. The author displayed results poorly and needs to be written better. Hence, the author needs to address the following comments before publishing this article in this journal. My comments are 

  1. The introduction needs to be precise. The author should improve the according to the present study hypothesis.
  2. The author should add novelty to the study. 
  3. The author should rewrite all the figure legends more clearly. 
  4. The author displayed many results, but they were not clearly written. So need to rewrite the results and discussion section.
  5. There needs to be more information about how the animal study was maintained and conducted. I mean ethical permission.
  6. Why did the author perform in male animals? In this kind of study, animal gender plays a central role. Explain.
  7. How did the author dose fixation for this study?
  8. The author has displayed Figure 8 in your previous publication. It should be removed from this manuscript or mentioned and appropriately cited. Also, It is very similar to the last publication. Poleschuk, T.S.; Sultanov, R.M.; Ermolenko, E.V.; Shulgina, L.V.; Kasyanov, S.P. Protective action of alkylglycerols under stress. STRESS 2019, 9, 1-8. https://doi.org/10.1080/10253890.2019.1660316

Author Response

Thank you for your comments. Answers are in the file below.

Round 2

Reviewer 1 Report (Previous Reviewer 2)

There were appropriate responses from the authors to the matters I raised.

One point

There is no group names in Fig 6.

Reviewer 3 Report (New Reviewer)

The authors have satisfactorily responded to all comments and made the necessary changes to the manuscript.

This manuscript is a resubmission of an earlier submission. The following is a list of the peer review reports and author responses from that submission.

Round 1

Reviewer 1 Report

In this study, authors evaluated the protective properties of alkylglycerols from marine sources against chronic stress in a rat model. The manuscript is very poorly written and not well organised. I think the manuscript is not ready for publication.

Some of the major issues are:

1)      The manuscript needs extensive language editing. The current version is totally unacceptable.

2)      There is no coherence in writing. For example, in the abstract, there is no background information.

3)      All the figures and figure legends are incomplete, e.g. titles are incomplete, no information about what the box plots are referring etc. The figures should be self-explanatory.

4)      It’s not clear how the AGs were quantified using GC-MS? What standards were used?

5)      Was the stress induction procedure a standard one? If so, it should be referenced.

Author Response

Dear Reviewer 

Thank you for your comments.

Our answer in file below.

Reviewer 2 Report

This paper describes the protective effect of marine alkylglycerol esters (AGs) on a rat chronic stress model. However, the data do not clearly support the protective effects of AGs on chronic stress in this animal model. The mechanisms of alleviative effects of AGs for abnormalities of lipid metabolism and the immune system have also not been adequately assessed.

Unfortunately, I think this manuscript is not worth publishing in Marine Drugs.

 Specific comments:

1. Although the authors only use non-parametric methods for statistical analysis of the data, parametric methods such as two-way ANOVA and multiple comparisons should also be used.

2. The authors calculated LDL-С and VLDL-С using the formulaof Friedwald [Ref 24], but this formula is intended for humans and is unlikely to be used for rats.

3. Authors should indicate the record of food intake, because metabolic profiles are largely influenced by intake amounts of diets.

4.Abstract, Ln16: “six weak” should be “six week”.

Author Response

Dear reviewere.

Thank you for your comments.

Our answer in file below.

Reviewer 3 Report

The manuscript by Dr. Sultanov et al describes their studies of the effects of oral alkylglycerols(AGs) on a rat model of chronic stress. The authors used a daily 2-hour exposure of immobilization to induce stress and determined the effects of AG on the biochemical and immune stress response. The experimental design consisted of four groups of rats: 1) unstressed control, 2) unstressed +AGs, 3) stressed, 4) stressed + AGs. The authors conclude that AGs protect stressed rats from the detrimental immune changes to chronic stress. The study design is appropriate and the methods are sound, but the manuscript is confusing to read, it needs clarification of many experimental results and conclusions, and correction of English usage. My specific comments are listed below.

1. The Abstract is too detailed. The sentences beginning with “Composition” through “different temperatures” (lines 11-15) can be deleted. I suggest adding section titles to the Abstract such as “Methods: The study was performed on…”. “Results: AGs at the selected dose…” “Conclusions: AGs prevent stress-induced…”. Line 26-27: replace “…improved lipid metabolism in rats” with specific lipid changes that account for improved lipid metabolism to be more specific. By my reading of the Results, the only statistically supported effects of AGs on stressed animals were an increase in blood lymphocytes (and neutrophil-lymphocyte ratio) and perhaps a decrease in triglycerides.

2. Line 31: Delete “definitively” which is too strong a term for the data presented. Line 24: delete “body weight” and replace with “body weight gain”.

3. After Section “2. Results” and before “Section 2.1”, the authors should summarize the experiment that was done. Section 2.1 abruptly begins to describe results before the experiment is actually described much later in the manuscript (Section 4.3.1).

4. It is confusing to describe the experimental groups in two different ways, as a number (groups 1-4) in some cases (all figures) and treatment descriptions “Control, AGs, Stress, AGs + Stress” in Tables 1, 3 and 4. The group numbers in all of the figures should be replaced with the treatment descriptions.

5. Table 1 is not placed in the proper order. It should be moved to before Figure 2.

6. Figure 1: This figure actually describes weight gain. The Y axis should be labeled as “Body Weight Gain (g)”.

7. Figure 1, Line 75: what does “Emissions” and “Point edge” mean?

8. There is no Table 2 in the manuscript. Tables should therefore be renumbered.

9. Section 2.2: Body weight changes under stress: The data is not graphically presented, but is summarized instead. The description is confusing. Perhaps a table showing the data would help clarify the text. The authors report that animals treated with AGs “did not lose weight” (line 101). What does Me=26.5 refer to? Is this “Mean weight gain” expressed in grams? If so, was the weight gain different in AG treated animals compared to controls? Also, please define “Me” or replace it with “Mean”.

10. Section 2.3, lines 110-113: Since there was no significant differences in erythrocytes, hematocrit and hemoglobin (lines 106-109, preceding paragraph), this paragraph (lines 110-113) should be deleted.

11. The terms granulocytes (line 122 and Appendix table) and neutrophils (Figure 5 line 128 and Appendix, neutrophil-lymphocyte ratio) seem to be used interchangeably. Granulocytes are not the same as neutrophils, so the use of these terms should be consistent.

12. I am bothered by several comments by the authors in Results and Discussion about some of their data, which show no statistical significance but they describe a “trend” in the data. They should be definitive. When there is no statistical significance, they should not confuse the reader by hedging their results by reporting trends. Their descriptions of “trends” in the data suggest that they 1) did not have sufficient numbers of animals in their groups (inadequate power), 2) did not use a proper dose of AGs to demonstrate statistical effects, 3) did not treat the animals long enough to demonstrate an effect, or 4) did not establish enough stress (which is possible since they did not demonstrate elevated cortisol or adrenal gland morphometry changes).

13. The authors found no statistical difference in the morphometry measurements of spleen, adrenal gland and thymus among the experimental groups (Section 2.4, lines 133-135). But they state that there was a “trend” in the adrenal gland index (Figure 6). I am not convinced by this observation. They should not confuse the reader and simply stay with their statistical conclusion.

14. Similarly, there were no statistical differences in testosterone or cortisol levels among the experimental groups. Lines 146-150 and Fig 7 should be deleted.

15. Materials and Methods: what was the age and sex of the animals used in this investigation. I only note in the Abstract that males were mentioned.

16. Line 135: Specifically report the method for lipid hydrolysis. Reference to Christie (ref 21) is not sufficient. Lipids can be hydrolyzed with acid or alkali, so which method was used?

17. Table 5. Typo: “himyl alcohol” should be “Chimyl alcohol”.

18. Line 311: What methodology was used in the Vector Kits to measure testosterone and cortisol? Were these immunoassays?

Author Response

Dear reviewer.

Thank you for your comments.

Our answer in the file below.